# Universal Connection through Art: Role of Mirror Neurons in Art Production and Reception

**DOI:** 10.3390/bs7020029

**Published:** 2017-05-05

**Authors:** Bartlomiej Piechowski-Jozwiak, François Boller, Julien Bogousslavsky

**Affiliations:** 1King’s College Hospital, Neuroscience Building, Denmark Hill, London SE5 9RS, UK; 2Department of Neurology, George Washington University Medical School, Washington, DC 20037, USA; fboller@mfa.gwu.edu; 3Clinique Valmont, Route de Valmont, 1823 Glion sur Montreux, Switzerland; jbogousslavsky@bluewin.ch

**Keywords:** art and brain, canonical neurons, art communication, universality

## Abstract

Art is defined as expression or application of human creative skill and imagination producing works to be appreciated primarily for their aesthetic value or emotional power. This definition encompasses two very important elements—the creation and reception of art—and by doing so it establishes a link, a dialogue between the artist and spectator. From the evolutionary biological perspective, activities need to have an immediate or remote effect on the population through improving survival, gene selection, and environmental adjustment, and this includes art. It may serve as a universal means of communication bypassing time, cultural, ethnic, and social differences. The neurological mechanisms of both art production and appreciation are researched by neuroscientists and discussed both in terms of healthy brain biology and complex neuronal networking perspectives. In this paper, we describe folk art and the issue of symbolic archetypes in psychoanalytic thought as well as offer neuronal mechanisms for art by emphasizing mirror/neurons and the role they play in it.

## 1. Introduction

Art transmits a universal message that can be appreciated despite cultural, religious, and economic differences across time. It has been attracting attention through generations and has served as a symbolic communicative system, a method of expression and abstracting. The main question concerns the role of art from a biological perspective. The other aspect is the neural circuitry and physiology underpinning this phenomenon. In this paper, we will focus on mirror neurons and the neuronal circuitry involved in the interaction between the artist and the viewer. 

## 2. Folk Art and Symbolic Art

Art has been practiced only by humans, and it was present in preliterate and prehistorical cultures. Art evolved with mankind with more and more sophisticated ways of expression matching ever-changing civilization [1]. It may be useful for the understanding of the neural underpinnings of art to consider so-called “primitive art”. According to Moore and Wilkinson [2], primitive art (a controversial term suggesting crudeness, incompetence, or ignorance will be referred to in the present paper as ‘folk art’) shows the very essence of human to human connection with its straightforward statement embedded in it. They consider its true simplicity as a beauty and its non-self-conscious virtue [2]:
“...The most striking quality common to all primitive art is its intense vitality. It is something made by people with a direct and immediate response to life. Sculpture and painting for them was not an activity of calculation or academism, but a channel for expressing powerful beliefs, hopes, and fears”.[2]


This direct, spontaneous expression which is directly transferable to the other person is the very core for further dissecting the neural mechanisms behind it. In the context of the early human social groups growing in size and complexity, and communicating through language and art the role of the latter may be considered important in promoting cohesion and survival [1,3]. In addition, folk art can offer a good model for gaining insights into the dialog between the artist and the receiver. Folk art is based on the use of symbols which reappear in many cultures or time periods baring similar significance and conveying similar message despite these geographic and historic barriers. One key in translating the universal message in symbolic art may be its reference and use of archetypes. The concept of the archetype in visual art was indirectly suggested by the psychoanalyst Carl Gustav Jung:
“An image can be considered archetypal when it can be shown to exist in the records of human history, in identical form and with the same meaning”.[4]

According to Stenudd [5], the Jungian archetypes carry meanings for the human mind to decipher and utilize in almost automated, subconscious way. He makes a reference to Jung’s theory on the presence of archetypes: [5]
“It seems to me that their origin can only be explained by assuming them to be deposits of the constantly repeated experiences of humanity”.[4]

This shows the recurrent pattern of humans using similar ways to communicate and express themselves in a symbolic way across the centuries. A good example of the subconscious or automated interpretation of symbolic art can be seen by looking at Figure 1.

Andrzejewski describes his production in this way:
“A form of the boat symbolizes human journey through life. It also symbolizes the cradle. It is the boat, which we wander through the world, which protects our home, bed, which gives us rest. In the end this is the boat of Haron, which takes us for the final journey to the other side of the Styx.”

Interpretation of any art piece is limited to the context in which it was created, that is to cultural/religious/historical background, and in the case of Haron’s boat it refers to Greek mythology, which was adopted into European culture starting with ancient Rome, and on through the Renaissance. Stephens lists a boat among Jung’s archetypes as a symbol of the body, carrying the self through life [6]. The main resource of images of archetypes and symbols is the Archive for Research in Archetypal Symbolism (ARAS). The conception of ARAS is linked to the Eranos Society established in August 1933 by Olga Fröbe-Kapteyn [7], Dutch origin photographer born in London. From the onset “Eranos Tagungen”—Eranos meetings would assemble many of Europe’s leading intellectuals to give scholarly lectures in the fields of religion, philosophy, history, art, and science. Of note is the fact that Carl Gustav Jung was an active member of this society for many years. Currently, ARAS contains more than 17,000 photographs of human creative artistic production, representing every culture over the past 30,000 years. ARAS is a unique resource for researchers allowing for exploration of human creativity across centuries [8]. 

## 3. Biological, Evolutionary, and Social Meaning of Art

In addition to the symbolic communication through artistic production the other property of art is group membership identification [1,3,9]. The informative and social properties of art, besides the aesthetic side of it, emerge 45,000–35,000 years ago at the time when art became a regular element of human existence [10]. For review and elaboration on the evolutionary gain of art see work by Zaidel at al. [11]. According to evolutionary theories, any brain function and reaction need to be translatable into biological gain such as survival. The main question concerns the relevance and meaning of aesthetics in art perception and also in biology [3]. There are studies confirming that infants as young as 2–3 months display beauty reaction to faces differing in race, gender, and age and this suggests that beauty reactions are present early in development. There is evidence that children as young as five years exhibit simple attractiveness aftereffects to faces [3,12]. Griffin and Langlois [13] showed that facial unattractiveness was a disadvantage, consistent with negativity bias but that beauty was good and positive. They found that unattractiveness negatively affected judgments of altruism intelligence. Generalizing to artworks, the two aspects of art—the production and the perception of it—should be taken into account as attractants for the receiver-content (embedded in symbols) and aesthetics.

## 4. Mirror Neurons and Viewers’ Reaction to Art: A Possible Relationship

One of the main questions concerns the neural systems that might be involved in art production and art reception. Both of these processes are active, and engaging the individual involved. The neural mechanisms involved in artistic activity have widely been discussed in the literature [10,14,15]. The recent discovery of the mirror neurons system may possibly shed additional light on the function of the brain in creating and appreciating art [16]. 

In laboratory work, Gallese and colleagues [17] investigated macaque monkeys (who in fact do not produce art) and found activation in the neurons in the rostral part of inferior area 6 in the frontal lobes (area F5) neurons not only when the monkeys performed predefined targeted actions such as grasping objects but also when they observed a similar action performed by the experimenter or other subjects including humans. In other words, witnessing action activates the same neural networks as those used to perform these activities. Based on these findings, Gallese and colleagues proposed the term ‘mirror neurons’ relating to mirroring of visual input into motor neuronal activities. In their later studies, Gallese and colleagues found a group of neurons in the posterior parietal area connected in a bidirectional way with the premotor cortex area F5 [18]. The importance of the discovery of mirror neurons and fronto-parietal reciprocal connections is that it demonstrates how the visual stimulus triggers primary motor pathways directly rather than relating to visual associative areas. Sensory features of the perceived actions are crucial to the activation of mirror neurons as they trigger the motor representation of the same action within the observer brain. This mirroring mechanism provides neurophysiological denominator for the primates to recognize actions of their peers. This communication is based on both the output side and the input side. Gallese and colleagues were able to show that sensory input other than visual, such as auditory, triggers the firing of a subgroup of mirror neurons as well. This suggests the existence of audiovisual mirror neurons and the possibility that the understanding of action is crucial to the activity of this neural network [19]. In other words, this cortical system pairs action observation with action execution, and by doing so enables individuals to ‘understand’ the behavior of others [18].

## 5. Anatomical Location of Mirroring Circuits in Humans

Similar neuronal networks were found in humans, where the observation of motor actions leads to activation of motor cortical areas [17,20]. The anatomical basis of mirror neurons in humans is the operculum of the inferior frontal gyrus and adjacent premotor cortex as well as the rostral part of the inferior parietal lobule [21]. In comparison to monkey mirror neurons network, the human system responds to a wider range of triggers and it may deploy in response to more subtle stimuli such as perception of a possible purposeful movement (e.g., a woman approaching an apple eventually to reach for it). The neurons firing in anticipation of a motor action (canonical neurons) are believed to work together with mirror neurons. That is to say that the human motor system codes both the goal of the motor action and the way it is executed. The ventral premotor cortex and inferior frontal gyrus (area 44) is suggested as the anatomical correlate to area F5 in the monkey [22]. The activation of canonical neurons was demonstrated in humans in response to objects such as sexual organs, tools, fruits, etc. The areas of the brain that were activated in response to these stimuli were not perceptive ones but the ones responsible for action generation [23,24].

Similar mirroring activity was demonstrated for emotional activities in monkeys and humans. The anatomical substrate in both species is insula. Its anterior part receives abundant connections from olfactory and gustatory systems as well as connections from the ventral part of the superior temporal sulcus where the facial recognition function is localized [25]. The human insular has a similar anatomical structure and it was shown to activate in response to olfactory and gustatory stimuli. Of interest is the fact that unpleasant odors activated the left ventral insular in right-handed individuals and the right ventral insular in those who were left-handed. These findings suggest lateralized processing of emotional odors as a function of handedness [26,27]. Other authors were able to demonstrate activation of the insular in response to the disgusted facial expression of others. A proportional correlation between the degree of facial disgust and the intensity of insular activation was demonstrated [28].

The insular area is the center responsible for experiencing disgust and recognizing it in other people. Wicker and colleagues found that both the exposure to disgusting odors as well as presentation of short clips of other people smelling the content of a glass and showing a facial expression of disgust led to activation of the anterior insular [29]. Other structures were shown to be activated in response to disgust, including the basal ganglia and anterior part of cingulate cortex [19]. 

Based on these findings, it can be postulated that the human brain is active in the first-person as well as in the third-person experience of motor actions and emotions. That is, when we see another person, an orchestra of neuronal activity commences as the meshwork of parietal and premotor neurons gets activated in the same way as if we were performing this activity ourselves. When we see someone’s face expressing aversion, our own insular structures get activated as if we ourselves were expressing a similar aversion.

Another group of researchers demonstrated that both the insular and anterior cingulated cortex both participate in mirroring effect for pain empathy. An interesting finding was that the amplitude of empathic brain response was related to the intensity of displayed emotion, the assessment of the fairness of the suffering person. This is a unique finding that may explain the ability to share and receive the others’ feelings [30].
“...we weep with the weeping, laugh with the laughing, and grieve with the grieving...”.[31]

This short passage from the famous work *De Pictura* (On Painting) of Leon Battista Alberti, the Renaissance humanist, may be a symbolic depiction of the mirroring neural network and its possible role in art creation and art appreciation. There are data from functional imaging studies suggesting the presence of tactile empathy. Keysers and colleagues [20], using functional Magnetic Resonance Imaging (fMRI) showed that the secondary cortex, namely fronto-parietal operculum extending to the lateral convexity of the inferior parietal lobule (especially upper lip of the lateral sulcus—SII/PV location), was activated in response to direct touch. The same pattern of activation was demonstrated when the participants observed someone or something else getting touched by tested objects [20]. If we use the example of Goya’s painting, *Desastres de la Guerra,* we can perhaps explain the feeling of tactile empathy. The image of human bodies dismembered, beheaded, torn, and tortured produces a bodily, empathetic sensation in the viewer.

Moreover, there are data suggesting that humans may develop an impression of movement based on a static image. Knoblich and colleagues [32] showed the presence of an action simulation assumption. They found that the more the actions one observes resemble the way one would carry them out, the more accurate the simulation is [32]. 

## 6. Mirroring Neural Circuitries and Art

Gallese [33] postulated a very interesting hypothesis of mirroring neurons linking action, perception, and cognition into one single interconnected domain. He suggested that first-person experience would not be different on the neuronal level than the third person one; both of these would be mirrored via the neuronal networks, thus both subjects would share the same functional state. This shared state involved two different bodies/minds, which are the subject of same functions, as ‘embodied simulation’. The latter would have neurobiological purpose of modelling the interactions of body/mind with surrounding environment. The mirroring system would be the biological mechanism for this interaction [33]. As mentioned above the input into mirroring system can be visual, tactile, auditory, and also emotional. As suggested by Gallese, the mirroring system may be involved in processing of meaning, based on these multimodal inputs. The illustrative example is coming from functional imaging studies when the same activation of premotor sectors for face, arm, or leg action was achieved via silent reading of or listening to words referring to these body parts. The same cortical areas would be activated during execution of motor activities or observation of these body parts. This view would support the role of mirroring neurons in understanding and mapping multisensorial inputs with action, which is key in sensory motor integration [33].

Moreover, Freedberg and Gallese [34] proposed that the activation of embodied mechanisms encompassing simulation of actions, emotions, and bodily sensations essential would constitute the basic frame of aesthetic response to art, not only artistic performance (live or recorded) but also stationary, e.g., images or other forms of visual art [34]. They further propose a theory of empathic response to art based on interaction of two elements. The first is defined as the association between the embodied simulation-triggered empathetic feelings in the art receiver and the content of the very art work. The second is the association between the above mentioned empathetic feelings of the art receiver and technique of artistic production including the very mechanics of art piece creation (chisel marks, texture, brush work, and so on) [34]. 

Abstract art neural activation is a puzzle within the framework of the mirror neurons explanation [35]. The activation would be based on induction of empathetic response/engagement in the art receiver/observer and igniting the simulation of respective motor program based on the visible creation marks as mentioned above. These marks would match the artist’s goal directed movements and hence would activate corresponding areas of the art receiver’s brain as part of the ‘embodied simulation’. Along these lines, Gallese suggested that motor simulation can be induced by static work of art, but not someone’s direct action. These complex relations between the artist and art receiver are graphically represented in Figure 2A–C.

## 7. Functional Imaging Data

In addition to the mirroring neuron circuitries, it is important to mention results of studies which utilized functional imaging in art appreciation/perception [15]. Kawabata and Zeki [37] investigated if there are areas of the brain that are specifically activated when viewing paintings that are considered beautiful. They used different categories of painting such as a portrait, a landscape, a still life, and an abstract composition. The participants in this study were able to view other paintings and judge them as beautiful, neutral, or ugly. Later, the same assessment was done but in the functional scanner. This study showed that the orbitofrontal cortex is differentially engaged during the perception of beautiful and ugly stimuli, regardless of painting type. Moreover, they showed that the mobilization of cortical activity was different for beautiful and ugly [37]. 

By contrast, Di Dio and colleagues [38,39] tested a very difficult hypothesis: whether the biological basis for the experience of beauty in art is subjective or objective. The authors used functional imaging techniques. They presented to non-expert viewers images of masterpieces of Classical and Renaissance sculpture. The images were either original or they contained a modified version of the same sculptures. The authors found that the observation of original sculptures, relative to the modified ones, produced activation of the right insular as well as of lateral occipital gyrus, precuneus, and prefrontal areas. When participants were asked to give an aesthetic judgment, the images judged as beautiful activated the right amygdala, relative to those judged as ugly. The authors concluded that the objective beauty perception is related to the insular region and the perception of subjective beauty was driven by one’s own emotional experiences and related to the activation of amygdala. That is, the insula was activated as part of a spontaneous reaction and the amygdala was activated as part of an induced aesthetic attitude [38,39].

## 8. Possible Role of Neurotransmitters

The role of neurotransmitters in art production can be gleaned from established artists with Parkinson’s Disease since their health depends on neurotransmitter medication [10]. One of the key questions concerns whether there is a representative neurotransmitter which can be linked with creative thinking and behavior. Dopamine has been proposed to be high on the list of candidate neurotransmitters by de Manzano et al. [40]. These authors studied the relationship between creative ability and dopamine D2 receptor expression in healthy individuals in laboratory settings. They measured D2 receptor densities with Positron Emission Tomography (PET) and radioligands and used standardized laboratory tests of divergent thinking which correlate positively with real-life creative activities, self-rated creativity, and objective measures of creative achievement. They found a negative correlation between divergent thinking scores and D2 density in the thalamus. Their results suggest that the D2 receptors, and specifically thalamic function, are important in creative activities, of the type measured in divergent thinking tests. De Manzano suggested that lower thalamic gating thresholds may increase thalamo-cortical information flow which may lead to increased performance on divergent thinking tests [40]. However, future studies aimed at understanding the relationship between neurotransmitters and art creativity could shed further light on this issue. 

## 9. Summary

Overall, these neurophysiological and functional imaging findings in humans of mirror, canonic neurons, multisensorial empathy, embodied simulation, and the role of neurotransmitters provide further understanding of biological/neuronal underpinnings of both art production and reception. The unique aspect of this neural circuitry is that the connection between the artist and the receiving viewer is based on the biological platform and in addition to this it appears to be universal [10]. This hypothesis aligns well with the view of the shared sense of being together with others in a single or unified experience [41]. In this context, it is the artistic experience that is in line with Gallese’s concept of multi-level connectedness and reciprocity among individuals [33,41]. These concepts fit with the notion of a connection between the artist and the receiver via multimodal sensory stimuli embedded in the art work.

It is worth noting that art is an important element of our civilization and the understanding of its neurological underpinnings has been stepped up and is actively being pursued in recent years (for review see, [15,35,42,43]). This scholarly area is of great interest for psychology, neuroscience, as well as clinical neurology since art is considered one of the treatment methods in brain disorders.

## Figures and Tables

**Figure 1 behavsci-07-00029-f001:**
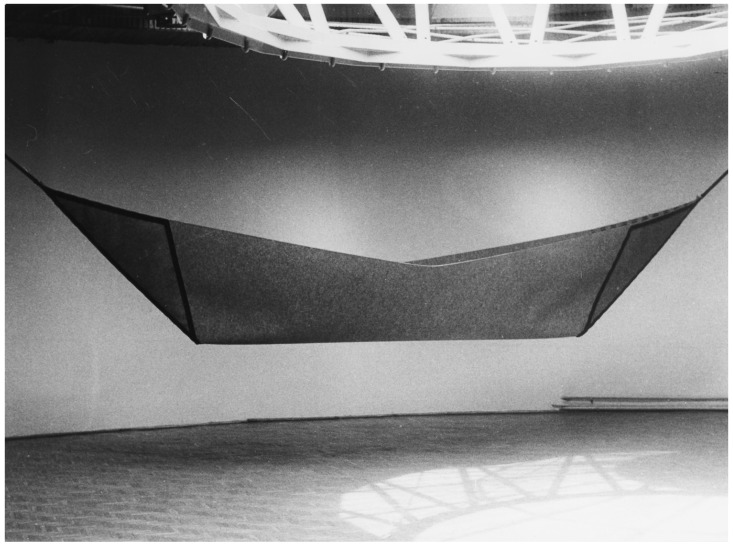
Stanisław Andrzejewski “Destiny” 1988, material: filt, leather, 140 × 900 cm.

**Figure 2 behavsci-07-00029-f002:**
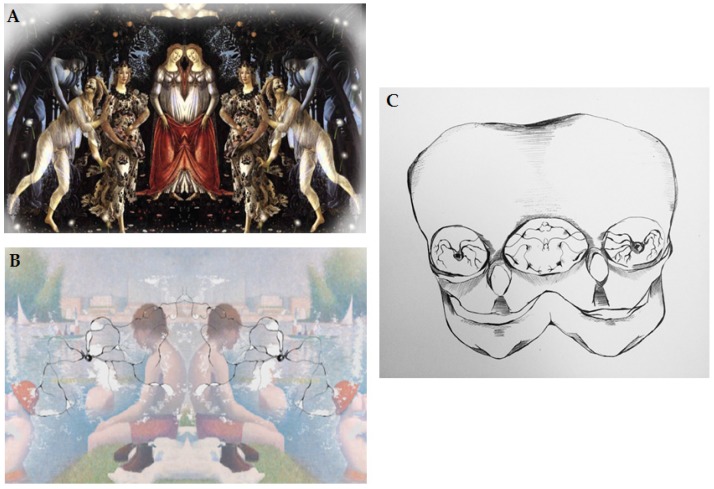
(**A**) Graphic depiction of reproduction of connection between the artist and art receiver; (**B**) Graphic depiction of mirror neurons and information transfer from the artist to the art receiver; (**C**) Graphic depiction of direct connection between the artist and art receiver via mirror neurons. From Julia Andrzejewska with permission [36].

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
