# Peer review of "Universal Connection through Art: Role of Mirror Neurons in Art Production and Reception"

_behavsci, 2017, doi:10.3390/bs7020029_

Round 1
Reviewer 1 Report
Piechowski-Jozwiak and colleagues revise the current literature in order to provide evidence for a key role of mirror/neuron systems in art production and reception. I feel this manuscript would be of interest for Behavioral Science, especially in the context of the Special Issue "Neuroscience of Art”.
I have only few minor concerns.
Page 4 lines 87/88: I would suggest the author to expand on the evolutionary gain of art.
Line 94: “art works” … maybe works of art? or artworks?
Page 6 line 212: “ “b ” this should be a typo.
I would suggest the Authors to group figure 2 in smaller panels within the same figure.
Author Response
Dear Reviewer, thank you very much for your suggestions. They were all executed and incorporated in the manuscript.
Reviewer 2 Report
Authors describe folk art and the issue of archetypes in psychoanalytic thought as well as
neuronal mechanisms focusing on the mirror neuron role.
I found the paper of interest and I have only minor concerns:
1. At the end of the manuscript Authors referred to neuropsychological studies supporting evidence about the neuronal mechanisms illustrated in the previous paragraphs. I suggest to the Authors to describe also clinical disorders (i.e., neurological and psychiatric disorders) to support the issue described along the manuscript. In my opinion, the adding of neuropsychological evidence may enrich and support the thesis advanced by the Authors.
Author Response
Dear Reviewer, thank you very much for the suggestion of incorporating disease related artistic performance. We discussed the design of this manuscript and we arrived at the conclusion that there are many papers, including the ones we authored, dedicated to artistic production in neurological disorders both in alteration of previous artistic styles and also development of new skills in artistic naive individuals as part of their disease. We deliberately decided no to include this into our manuscript for two reasons. First of all we wanted to concentrate on the neural genesis of the art from the historical, neurological and neuropsychological perspective to reflect the normal brain function in the context of artistic production. The second reason for not incorporating disease themes here is logistical due to limitation of the word count of the paper. I do hope that you would accept this explannation and allow for publicaiton in this form.
Sincerely yours.